# Biological Activities and Biochemical Composition of Endemic *Achillea fraasii*

**DOI:** 10.3390/microorganisms11040978

**Published:** 2023-04-09

**Authors:** Yagmur Tunca-Pinarli, Atakan Benek, Dilay Turu, Mustafa Eray Bozyel, Kerem Canli, Ergin Murat Altuner

**Affiliations:** 1Department of Biology, Graduate School of Natural and Applied Sciences, Dokuz Eylül University, Izmir 35390, Türkiye; 2Department of Biology, Graduate School of Natural and Applied Sciences, Kastamonu University, Kastamonu 37150, Türkiye; 3Department of Biology, Faculty of Arts and Sciences, Çanakkale Onsekiz Mart University, Çanakkale 17020, Türkiye; 4Department of Biology, Faculty of Science, Dokuz Eylül University, Izmir 35390, Türkiye; 5Fauna and Flora Research and Application Center, Dokuz Eylül University, Izmir 35390, Türkiye; 6Department of Biology, Faculty of Science, Kastamonu University, Kastamonu 37150, Türkiye

**Keywords:** *Achillea fraasii*, antibacterial activity, antifungal activity, antioxidant activity, antibiofilm activity, GC-MS

## Abstract

In this study, we investigated the antimicrobial, antioxidant, and antibiofilm activities and the biochemical composition of *Achillea fraasii*. The antimicrobial activity of *A. fraasii* ethanol extract (AFEt) was tested against 48 strains, and this is the first study testing the antimicrobial activity of this plant to this extent. The antioxidant activity was determined using the DPPH assay, and the antibiofilm activity of *A. fraasii* aqueous extract (AFAq) against five strains was assessed. The chemical composition of the plant extract was determined using GC-MS with artemisia ketone (19.41%) as the main component. The findings indicated that AFEt displayed antimicrobial activity against 38 strains, with a particular efficacy observed against various *Staphylococcus aureus* strains, such as *S. aureus* ATCC 25923, clinically isolated, multidrug resistant (MDR), and methicillin-resistant (MRSA) strains. In addition, the highest activity was observed against *Enterococcus faecium*. Moreover, the extract demonstrated activity against *Candida* strains. The plant extract also showed relatively good antioxidant activity compared to ascorbic acid, with an *EC*_50_ value of 55.52 µg/mL. However, AFAq acted as a biofilm activator against *Escherichia coli* ATCC 25922, increasing the biofilm formation by 2.63-fold. In conclusion, our study demonstrates the potential of *A. fraasii* as a source of antimicrobial and antioxidant agents.

## 1. Introduction

The rise of multidrug resistance (MDR) is a major global health concern and has led to significant challenges in the treatment of infectious diseases [1]. MDR often occurs due to the overuse and misuse of antibiotics, which can give rise to the selection and spread of resistant strains. The World Health Organization (WHO) predicts that resistance to antibiotics will pose a severe risk to public health within the twenty-first century [2].

One of the mechanisms that can cause antibiotic resistance is the formation of biofilms. Biofilm is a matrix in which bacteria are embedded and is extremely resistant to external stresses such as UV light, chemical biocides, host immunological responses, and antibiotics [3,4]. Biofilm development is the primary virulence factor of many bacteria that cause chronic infectious diseases [5]. The current antimicrobial treatment methods and materials often fail to remove the biofilm structure at the site of infection [6]. 

The declining efficacy of antibiotics and the discovery of new side effects have made it necessary for scientists to explore hit compounds with new antimicrobial and antibiofilm effects. The development of new antibiotics and antibiofilm agents is crucial for our ability to combat MDR and biofilm formations, which provide effective treatment options for infectious diseases [7].

Biofilms can have advantageous effects based on their compositions and locations, notwithstanding their unfavorable effects. In some cases, biofilms can be used for industrial and medical purposes, such as wastewater treatment and bioremediation. Therefore, in certain cases, the induction of biofilm formation has favorable applications [8,9].

Several studies have shown that antioxidant compounds can enhance the efficacy of antibiotics in treating infections [10]. These compounds are also used against oxidative stress that directly or indirectly damages various organs and contributes to the development of numerous health problems, such as cancer, DM (diabetes mellitus), neurodegenerative diseases, CVD (cardiovascular disease), and atherosclerosis. In addition to being related to the effects of antibiotics, antioxidant compounds inhibit free radicals that can damage the cell structure and convert them into molecules without toxic effects [11,12].

Many studies in the literature have revealed the potential use of plants for the discovery of new compounds with antibiotic, antioxidant, and antibiofilm properties. 

The WHO has shown that traditional medicines derived from medicinal plants are still beneficial in 80% of developing countries. In total, it is believed that there are around 374,000 medicinal plant species, of which 28,000 are known to have extensive applications as complementary and alternative medicines. In addition, WHO defines more than 20,000 medicinal plant species as potential sources of new drugs [13,14]. In addition to the 30,000 antimicrobial molecules that have been extracted from plants, there are more than 1340 plants with known antibacterial activities [14].

*Achillea fraasii* is an endemic species of the Asteraceae family. It is native to southeastern Europe and Türkiye. Its most characteristic features are its long inflorescences and woolly leaves. The plant’s stem is 10–15 cm long, silky, woolly, cylindrical, and a rhizome. The flowering period is in June and July, and it grows on calcareous soils at an altitude of 1500 m [15].

*Achillea fraasii* is a plant that has been shown to possess antimicrobial properties, rendering it an attractive candidate for research on its potential applications in medicine. According to the literature, *Achillea fraasii* contains several compounds, such as flavonoids, tannins, and sesquiterpene lactones, that have been shown to exhibit antibacterial and antifungal activities. In addition, *Achillea fraasii* has also been found to have antioxidant and anti-inflammatory properties, which could be beneficial for the treatment of infections and related conditions. Given its potential as a source of natural antimicrobial agents, *Achillea fraasii* is a promising subject for further research on the mechanisms of its antimicrobial activity and its potential use in the development of new therapies for infectious diseases [16,17,18].

In this study, *A. fraasii* ethanol and aqueous extracts were prepared, and the antibacterial, antifungal, antioxidant, and antibiofilm activities and biochemical composition of the plant were determined. Although there are several studies in the literature regarding *A. fraasii*, our study is the first study testing antimicrobial activity of this plant to this extent.

## 2. Materials and Methods

### 2.1. Endemic Plant Samples

*Achillea fraasii* Sch.Bip. (Syn: *Achillea fraasii* var. *troiana* Aschers. & Heilmerl) was used in this study. It was collected from Çanakkale Kazdağı (Mount Ida) and identified by Dr. Mustafa Eray Bozyel. The herbarium voucher specimen was deposited at FAMER (Fauna and Flora Research and Application Center), Dokuz Eylül University (with the following personal herbarium specimen number: FFDEU-ERB0117).

### 2.2. Microorganisms and Inoculum Preparation

A total of 48 strains were used in this study, including 16 standard, 11 clinically isolated, 11 multidrug-resistant, 7 food-isolated bacterial, 2 clinically isolated, and 1 standard fungal. The bacterial strains were incubated for 24 h at 37 °C, and the fungal strains were incubated for 48 h at 27 °C. To standardize the inoculations containing approximately 10^8^ cfu.mL^−1^ of the bacterial strains and approximately 10^7^ cfu.mL^−1^ of the fungal strains, they were set to the 0.5 McFarland standard in sterile 0.9% NaCl solution. 

### 2.3. Extraction Method

After *A. fraasii* was identified and dried, 100 g of the whole plant sample was ground in a blender. After grinding, the powdered sample was shaken in 200 mL of pure ethyl alcohol (Sigma-Aldrich, Saint Louis, MO, USA) for 48 h at 160 rpm at room temperature. After 48 h, the *A. fraasii* ethanol extract (AFEt) was filtered into glass balloons using Whatman No 1. filter paper. Ethyl alcohol was then evaporated at 30 °C using a rotary evaporator Buchi R3 (BÜCHI, Labortechnik AG, Postfach, Flawil, Switzerland), in a vacuum. The remnants of the extract in the balloon were collected and dissolved in ethyl alcohol to prepare the ethyl alcohol extract stock solution.

The *A. fraasii* aqueous extract (AFAq) to be used in the antibiofilm test was obtained in the same manner as the ethyl alcohol extract. When the filtering process was completed, the extracts were frozen at −80 °C for 24 h. The frozen plant extracts were attached to a freeze dryer, and the water in the sample was removed. The remnants of the extract were used to prepare the aqueous extract stock solution by adding distilled water. The stock solution was sterilized through a 45 µm filter.

AFEt was used in antibacterial, antifungal, and antioxidant tests and to determine the biochemical composition of the extract. Since this extract contains ethanol, it was replaced with AFAq in order not to inhibit the growth of bacteria in the antibiofilm study. 

### 2.4. Antibacterial and Antifungal Activity Test

Using Andrews’ disk diffusion method, the antibacterial and antifungal effects of the AFEt were assessed [16]. A total of 50, 100, and 200 µL (3.77 mg, 7.55 mg, and 15.10 mg, respectively) of AFEt were loaded onto Oxoid Antimicrobial Susceptibility Test Disks with a 6 mm radius. To evaporate the ethyl alcohol in the extract, which could alter the test results, the disks were left to dry overnight at 30 °C under sterile conditions. After drying, standardized microorganisms were inoculated onto Petri dishes containing Mueller Hinton Agar (BD Difco, East Rutherford, NJ, USA). The extract-loaded disks were placed on the culture media, and the Petri dishes were incubated according to the time and temperature combinations reported in Section 2.2 on the inoculum preparation. At the end of the incubation period, the diameters of the inhibition zones formed around the disks were measured in mm and recorded. In this study, sterile blank disks and ethyl-alcohol-loaded disks were used as negative controls. As positive controls, we used Gentamicin (Gen) for Gram-negative bacterial strains, Ampicillin (Amp) for Gram-positive bacterial strains, and Tobramycin (Tob) for fungal strains in order to compare the results obtained. 

### 2.5. Antioxidant Activity Test

The DPPH method is based on the assessment of the DPPH radical scavenging properties of antioxidant compounds found in plant extracts. To prepare the DPPH solution, 0.0039 g of 2,2-diphenyl-1-picrylhydrazyl (DPPH) was added to 50 mL of ethanol, and it was kept in the dark until further use [19]. A 96-well plate containing DPPH solution and different concentrations of AFEt ranging between 1.075 and 200 µg/mL was then incubated at room temperature for 30 min in the dark. A plate reader (Biotek Microplate Spectrophotometer, Winooski, VT, USA) was used to measure the absorbances of the wells at 515 nm after the incubation time. In this study, ascorbic acid was used as the positive control. 

The *EC*_50_ value was calculated according to the following equation:$$Y=Min+\frac{Max-Min}{1+{\left(\frac{X}{EC_{50}}\right)}^{Hill\;coefficient}}$$

### 2.6. Antibiofilm Activity Test

The method previously proposed by Karaca et al. [20] was used with modifications in the antibiofilm activity test. This method has two stages, namely, the determination of the conditions for biofilm formation and determination of the antibiofilm activity of AFAq.

Determination of biofilm formation conditions: To determine the antibiofilm activity of the AFAq, five bacterial strains were selected: *Escherichia coli* (clinical-isolated), *Listeria innocua* (food-isolated), *E. coli* ATCC 25922, *Listeria monocytogenes* ATCC 7644, and *Bacillus subtilis* DSMZ 1971. All bacterial strains were adjusted to 0.5 McFarland, transferred to microplates containing TSB supported by 0.0%, 0.5%, 1.5%, 2.0%, and 2.5% glucose, and cultured for 24 to 48 h at 37 °C.

After the incubation period, 200 µL of crystal violet was transferred into each well and incubated for 15 min before being rinsed with distilled water (dH_2_O). The crystal violet was drained after the incubation period, and all the wells were then cleaned with dH_2_O once again. Finally, 200 µL of ethyl alcohol: acetone (30:70% (*v*/*v*)) solution was pipetted into all the wells, which were then incubated for 15 min. At the end of the incubation period, the ethyl alcohol: acetone solution was collected from each well and transferred to the wells of clean microplates. The absorbance of each well was determined at 550 nm. As a result, the optimal parameters for biofilm formation were obtained as 48 h of incubation time and a 1.5% glucose concentration for all the strains used in the test.

Determination of antibiofilm activity: A concentration range of AFAq was tested for its antibiofilm potential against the five bacterial strains used in the first step. The same steps carried out in the first part of the test were followed, but the bacteria were only incubated for 48 h at 37 °C in TSB containing 1.5% glucose. The antibiofilm potentials of different concentrations of AFAq were determined at 550 nm.

### 2.7. Gas Chromatography–Mass Spectroscopy Method (GC-MS)

A gas chromatography–mass spectrometry device is used to separate the compounds present in a solution, and mass spectroscopy is used to structurally define the compounds [21]. In this study, the Agilent 8890 GC-MS instrument was used. The injector temperature was 350 °C, and He gas was used as the carrier (1 mL/min). The injector mode is 10:1 split, and the injector volume is 1 microliter. The furnace temperature was increased from 40 °C to 150 °C at an increment of 4 degrees per minute, from 150 °C to 180 °C at 3 degrees per minute, from 180 °C to 230 °C at 2 degrees per minute, and from 230 °C to 280 °C at 1 degree per minute. The ions formed as a result of electron ionization by the GC-MS technique were separated according to their mass/charge ratios and recorded with the detector. The compounds were determined by cross-matching the data of the compounds in the most recent Nist and Wiley data libraries. 

### 2.8. Statistics

All disk diffusion tests for antimicrobial and antifungal activities were performed in triplicate. The Pearson correlation coefficient was calculated to represent any correlation between the amount and biological activity of the extract, and the ANOVA test was used to determine whether the differences between the replicates were statistically significant or not. For the statistical analysis, R Studio 3.3.2 was used.

## 3. Results

### 3.1. Antibacterial and Antifungal Activities of AFEt

The data obtained from the study of the inhibition zone diameters are shown in Table 1. The negative controls showed no activity, and the difference between parallels was insignificant (*p* > 0.05) in the antimicrobial activity tests. The difference between the results for the 50, 100, and 200 µL applications was also insignificant (*p* = 0.751). A very weak positive correlation was found between the increase in the AFEt concentration and its antimicrobial activity (Pearson correlation coefficient = 0.055).

AFEt demonstrated antimicrobial activity against 38 out of the 48 strains (Table 1). Among them, three showed high susceptibility (≥15 mm), twenty showed moderate susceptibility (10–14 mm), and fifteen showed low susceptibility (7–9 mm) [2]. Notably, all the strains of *S. aureus* were susceptible to AFEt, and the most susceptible Gram-positive bacterial strain was FI2, with a 28 mm inhibition zone, being higher than those for Gentamicin and Tobramycin. The most susceptible Gram-negative bacterial strain was MDR11, with a 13 mm inhibition zone, with AFEt exhibiting a higher activity than all the antibiotics. All the fungal strains were susceptible to AFEt, with CI12 and CI13 exhibiting more effective results than the positive controls.

### 3.2. Antioxidant Activity of AFEt

The results for the antioxidant activity of AFEt and ascorbic acid are provided in Table 2. According to the results, AFEt showed lower antioxidant activity than the positive control, ascorbic acid. The *EC*_50_ value of AFEt, determined to be 55.52 µg/mL, represents the concentration at which it scavenges 50% of the DPPH radicals.

The statistical analysis of the DPPH test results for the AFEt extract indicates no significant difference between the test replicates (*p* = 0.9925). In addition, the correlation test suggests a strong negative correlation between the increase in the extract concentration and DPPH absorbance (Pearson correlation coefficient = −0.9319768). This strongly negative correlation suggests that as the extract concentration increases, DPPH radical scavenging activity also increases, leading to a decrease in DPPH absorbance. Consequently, this finding highlights the dose-dependent antioxidant activity of the AFEt extract.

### 3.3. Antibiofilm Activity of AFAq

The effects of AFAq on biofilm formation among the tested microorganisms were investigated in the study, and the fold increases in biofilm formation were calculated (Table 3).

According to the fold increases in biofilm, it can be observed that the highest fold increase was observed in *E. coli* ATCC 25922, being 2.63.

### 3.4. Analysis of the Biochemical Composition of AFEt

The biochemical composition of AFEt and its percentages are given in Table 4, according to the data obtained from the GC-MS analysis. 

According to Table 4, artemisia ketone (19.41%), aromadendrene (5.11%), and (R)-lavandulyl (R)-2-methylbutanoate (4.86%) are major compounds in the biochemical composition of AFEt.

## 4. Discussion

AFEt was tested against 48 strains to determine its antibacterial and antifungal activities and showed activity against 38 strains. It is possible that the antimicrobial activity of AFEt is produced by some major compounds found in the extract, such as artemisia ketone, aromadendrene, copaene, ledol, neophytadiene, pentadecane, and heneicosane, which were revealed in the results of the GC-MS analysis and whose antibacterial and antifungal activities were determined in previous studies.

The antibacterial activity results of the standard, clinically isolated, multidrug-resistant (MDR), and methicillin-resistant (MRSA) *S. aureus* strains used in the study were remarkable. *S. aureus* is a bacterium that causes various organ and tissue abscesses; skin, urinary tract, lung, and central nervous system infections; endocarditis; and osteomyelitis diseases. Therefore, antimicrobial activity against *S. aureus* is essential, since this bacterium increases the number of hospital deaths and is a severe public health threat. Compounds with antibacterial activity, as determined by the results of GC-MS analysis, support the antibacterial effect of *A. fraasii* [32,33].

The results of this study showed that AFEt exhibited antifungal activity against the three *Candida* strains tested. The antifungal activity of AFEt may be due to the presence of artemisia ketone, which has previously been shown to be an antifungal agent [28]. Fungal infections caused by *Candida* strains continue to be a major health problem, resulting in high mortality rates [34]. Mortality rates remain high even when antifungal drugs are used against *Candida* infections, which results in expensive medical costs for patients and governments. The antifungal activity exhibited by *A. fraasii* is important in today’s world, in which the need for antifungal agents is increasing [35].

Magiatis et al. [17] previously tested the essential oils of *A. fraasii* against *C. albicans* ATCC 10231, *C. glabrata* ATCC 28838, *C. tropicalis* ATCC 13801, *E. coli* ATCC 25922, *Enterobacter cloacae* ATCC 13047, *K. pneumoniae* ATCC 13883, *P. aeruginosa* ATCC 227853, *S. aureus* ATCC 25923, and *S. epidermidis* ATCC 12228 by minimum inhibitory concentration (MIC) tests. As a result, they observed that the essential oils of *A. fraasii* presented antibacterial activities against all the bacteria tested, with MIC values ranging between 3.21 and 6.87 mg/mL. When these results are compared to our results, it can be observed that only the results regarding *S. epidermidis* are different in terms of the common bacteria used in these two studies. On the other hand, AFEt exhibited antifungal activity against *C. albicans*, *C. glabrata,* and *C. tropicalis*, whereas the essential oils of *A. fraasii* did not show any antifungal activities against these yeast strains. The possible reasons for these differences may be based on both the strains and extracts used in these two studies.

In addition, there are several studies on the antibacterial activities of ethanol extracts from various *Achillea* species. For instance, Baris et al. [36] investigated the antibacterial activities of three *Achillea* species, namely, *Achillea aleppica* subsp. *aleppica* (AA), *Achillea aleppica* subsp. *zederbaueri* (AZ), and *Achillea biebersteinii* (AB), against *C. albicans*, *E. aerogenes*, *E. cloacae* ATCC 23355, *E. coli* ATCC 25922, *Klebsiella oxytoca*, *K. pneumoniae* ATCC 13883, *P. aeruginosa* ATCC 27853, *S. typhimurium* ATCC 14028, *S. aureus*, *S. epidermidis* ATCC 12228, and *Streptococcus pyogenes*. The results obtained in these two studies were similar, excepting those for *E. aerogenes*, *K. pneumoniae*, and *S. epidermidis*. The differences between the results may be attributed to the different strains and plants used in the two studies.

In comparison to the *Achillea aleppica* subsp. *aleppica* investigated by Çolak et al. [37], our study focused on a different species within the same genus. Çolak et al. observed a weaker antimicrobial effect against *E. faecium* relative to our findings, while reporting stronger effects against *S. aureus* strains. The observed discrepancies in antimicrobial activities might be attributed to both the differences in plant species and the choice of solvents employed.

*E. faecium* (FI) demonstrated a significant antimicrobial effect in our study, with zone diameters of 20 mm for 50 µL, 26 mm for 100 µL, and 28 mm for 200 µL. This finding is particularly important considering the role of *E. faecium* in healthcare-associated infections (HAIs). According to the World Health Organization (WHO), HAIs affect 3.5% to 12% of hospitalized patients in high-income countries and result in excess deaths [38]. *E. faecium* is a major cause of infection in healthcare settings, and vancomycin-resistant *E. faecium* (VREfm) poses a serious threat to immunocompromised patients [39]. The high prevalence of vancomycin resistance in *E. faecium* clinical isolates in countries like Australia highlights the need for alternative treatment options. Our findings of strong antimicrobial activity against *E. faecium* (FI) may contribute to the development of new therapeutic options for combating HAIs caused by this pathogen.

In general, it is expected that an increase in extract concentration would lead to larger zones of inhibition, indicating a stronger antimicrobial activity. However, in our study, some exceptions were observed for certain strains, including *E. coli* ATCC 25922, *S. boydii* (CI), *C. glabrata* (CI), *E. coli* (MDR), *K. pneumoniae* (MDR), and *P. rustigianii* (MDR), where smaller or no zones of inhibition were seen at higher levels of added extract. This phenomenon may be attributed to the presence of multiple compounds within the extract, which can exhibit different mechanisms of action and may interact either synergistically or antagonistically. It is hypothesized that the decrease in antimicrobial activity despite the increase in extract concentration may be due to the occurrence of an antagonistic effect, where one or more compounds within the extract suppress the antimicrobial activity of other components. Further research into the individual compounds and their interactions is needed to confirm this hypothesis and elucidate the mechanisms underlying these observations.

Living organisms are constantly exposed to reactive oxygen species generated as a result of respiratory, metabolic, or disease stress [40]. It is important to eliminate oxidation caused by reactive oxygen species, which cause many diseases, and to neutralize free radicals [41]. As a result of the antioxidant activity test, AFEt showed relatively strong antioxidant activity compared to ascorbic acid, which was used as a positive control, with the *EC*_50_ at 55.52 µg/mL. It is believed that this effect may be due to the copaene and neophytadiene compounds in the chemical content of the plant, the effects of which have been demonstrated previously [25,32]. Baris et al. [39] also tested the DPPH scavenging potentials of AA, AZ, and AB and identified *EC*_50_ values of 33, 33, and 32 μg/mL, respectively. The variations observed in the results are due to differences between the plant samples used in the two studies.

As a result of the data obtained from the antibiofilm test, it was determined that the plant extract did not exhibit antibiofilm activity against any of the microorganisms used in the assay. Contrary to expectations, AFAq was found to act as an activator of biofilm formation in *E. coli* ATCC 25922, increasing the biofilm formation by 2.63-fold. Samoilova et al. [42] previously proved that water extracts of *Zea mays*, *Betula pendula*, *Tilia cordata*, *Vaccinium vitis-idaea*, and *Arctostaphylos uva-ursi* and low concentrations of *Camellia sinensis* stimulate biofilm formation in *E. coli*, with an increase of up to 3-fold. They also proposed that the biofilm-triggering activity of these extracts may be due to their pro-oxidant properties. Nevertheless, further research is still needed in order to understand the biofilm activator activity of the extract in the *E. coli* ATCC 25922 strain.

As a result of GC-MS analysis, twenty-two different compounds were identified in AFEt. The biological activities of eight of these compounds had been demonstrated in previous studies. It was seen that the compounds with known biological activities support the antibacterial, antifungal, and antioxidant activities of AFEt obtained as a result of this study. Magiatis et al. [17] also determined the major constituents of the essential oils of *A. fraasii* by GC-MS, but the compounds determined in this study were completely different from the compounds determined in our study. The observed differences can primarily be attributed to the different extracts used in these two studies.

To investigate the biological activities of the primary compounds in AFEt and whether AFEt exhibits a more potent antibacterial effect than the sum of its individual components, we recommended analyzing the synergistic effects of the main compounds. This will help to elucidate the potential synergy between different compounds in AFEt and provide insight into their combined mechanism of action.

## 5. Conclusions

As a result of this study, it was observed that AFEt showed antibacterial, antifungal, and antioxidant activities, whereas AFaq exhibited biofilm-triggering activity. Biofilms can have both beneficial and detrimental effects depending on their location and composition. In some cases, biofilms can be useful for industrial and medical purposes, such as wastewater treatment, bioremediation, and medical implants. For example, biofilms can help to degrade organic pollutants in wastewater and can provide a stable surface for the growth of beneficial bacteria on medical implants. However, in other cases, biofilms can be harmful, as in the formation of dental plaque, chronic infections, and biofouling in water systems.

Therefore, purification and further studies are required to determine whether the antibacterial, antifungal, and antioxidant activities are due to a single known biochemical compound, such as artemisia ketone, aromadendrene, or copaene, or to a synergistic effect.

## Figures and Tables

**Table 1 microorganisms-11-00978-t001:** Disk diffusion test results of AFEt (inhibition zone diameters in mm).

No	Microorganisms	50 **µL ^1^**	100 **µL ^1^**	200 **µL ^1^**	Gen	Amp	Tob
1	*Bacillus subtilis* DSMZ 1971	10.00 ± 0.00	12.00 ± 0.00	12.00 ± 0.50	30	41	26
2	*Candida albicans* DSMZ 1386	0.00 ± 0.00	10.00 ± 0.00	10.00 ± 0.50	12	0	13
3	*Enterobacter aerogenes* ATCC 13048	0.00 ± 0.00	0.00 ± 0.00	0.00 ± 0.00	24	0	18
4	*Enterococcus faecalis* ATCC 29212	8.00 ± 0.00	9.00 ± 0.00	10.00 ± 0.00	12	14	8
5	*Escherichia coli* ATCC 25922	8.00 ± 0.00	8.00 ± 0.00	0.00 ± 0.00	22	6	20
6	*Listeria monocytogenes* ATCC 7644	11.00 ± 0.00	13.00 ± 0.00	15.00 ± 0.00	28	23	24
7	*Pseudomonas aeruginosa* DSMZ 50071	8.00 ± 0.00	10.00 ± 0.50	12.00 ± 0.00	15	0	22
8	*Pseudomonas fluorescens* P1	8.00 ± 0.00	8.00 ± 0.50	12.00 ± 0.00	13	14	12
9	*Salmonella enteritidis* ATCC 13076	0.00 ± 0.00	0.00 ± 0.00	0.00 ± 0.00	21	16	15
10	*Salmonella typhimurium* SL 1344	10.00 ± 0.00	11.00 ± 0.00	12.00 ± 0.00	24	13	15
11	*Staphylococcus aureus* ATCC 25923	10.00 ± 0.00	11.00 ± 0.00	13.00 ± 0.00	21	25	14
12	*Staphylococcus epidermidis* DSMZ 20044	0.00 ± 0.00	0.00 ± 0.00	0.00 ± 0.00	22	24	20
13	*Staphylococcus hominis* ATCC 27844	9.00 ± 0.00	11.00 ± 0.00	13.00 ± 0.00	18	0	16
14	*Staphylococcus warneri* ATCC 27836	9.00 ± 0.50	11.00 ± 0.00	13.00 ± 0.00	23	0	18
15	*Bacillus cereus* RSKK 863	8.00 ± 0.50	12.00 ± 0.00	14.00 ± 0.00	24	0	18
16	*Shigella flexneri* RSKK 184	10.00 ± 0.00	11.00 ± 0.00	11.00 ± 0.50	18	0	17
17	*Acinetobacter baumannii* CECT 9111	8.00 ± 0.50	10.00 ± 0.00	11.00 ± 0.00	13	0	22
18	*Enterococcus durans* (FI)	0.00 ± 0.00	0.00 ± 0.00	0.00 ± 0.00	11	28	13
19	*Enterococcus faecium* (FI)	20.00 ± 0.00	26.00 ± 0.00	28.00 ± 0.00	28	32	15
20	*Klebsiella pneumoniae* (FI)	7.00 ± 0.00	7.00 ± 0.00	8.00 ± 0.00	19	6	23
21	*Listeria innocua* (FI)	0.00 ± 0.00	0.00 ± 0.00	0.00 ± 0.00	13	13	15
22	*Salmonella infantis* (FI)	0.00 ± 0.00	8.00 ± 0.00	8.00 ± 0.00	17	14	14
23	*Salmonella kentucky* (FI)	0.00 ± 0.00	0.00 ± 0.00	0.00 ± 0.00	12	15	16
24	*Escherichia coli* (FI)	0.00 ± 0.00	0.00 ± 0.00	0.00 ± 0.00	20	0	0
25	*Staphylococcus aureus* (CI)	12.00 ± 0.00	13.00 ± 0.00	15.00 ± 0.00	22	0	18
26	*Staphylococcus mutans* (CI)	9.00 ± 0.00	10.00 ± 0.00	10.00 ± 0.50	22	0	24
27	*Staphylococcus hominis* (CI)	8.00 ± 0.00	9.00 ± 0.00	10.00 ± 0.00	9	26	11
28	*Staphylococcus haemolyticus* (CI)	7.00 ± 0.00	8.00 ± 0.00	8.00 ± 0.00	10	0	10
29	*Staphylococcus lugdunensis* (CI)	8.00 ± 0.00	8.00 ± 0.00	8.00 ± 0.00	17	8	18
30	*Shigella boydii* (CI)	8.00 ± 0.00	0.00 ± 0.00	0.00 ± 0.00	20	0	18
31	*Acinetobacter baumannii* (CI)	8.00 ± 0.00	8.00 ± 0.00	8.00 ± 0.00	18	0	16
32	*Shigella flexneri* (CI)	7.00 ± 0.00	8.00 ± 0.00	8.00 ± 0.00	16	23	14
33	*Staphylococcus aureus* (CI)	7.00 ± 0.00	7.00 ± 0.00	7.00 ± 0.00	22	17	16
34	*Enterococcus faecalis* (CI)	9.00 ± 0.00	9.00 ± 0.00	10.00 ± 0.00	12	8	10
35	*Klebsiella pneumoniae* (CI)	8.00 ± 0.00	8.00 ± 0.00	8.00 ± 0.00	18	8	18
36	*Candida tropicalis* (CI)	0.00 ± 0.00	7.00 ± 0.00	8.00 ± 0.00	0	0	0
37	*Candida glabrata* (CI)	10.00 ± 0.00	10.00 ± 0.00	0.00 ± 0.00	7	0	8
38	*Escherichia coli* (MDR)	8.00 ± 0.00	8.00 ± 0.00	0.00 ± 0.00	8	0	9
39	*Klebsiella pneumoniae* (MDR)	8.00 ± 0.00	8.00 ± 0.00	0.00 ± 0.00	15	8	20
40	*Acinetobacter baumannii* (MDR)	10.00 ± 0.00	10.00 ± 0.00	10.00 ± 0.00	0	0	0
41	*Enterobacter aerogenes* (MDR)	0.00 ± 0.00	0.00 ± 0.00	0.00 ± 0.00	16	0	18
42	*Serratia odorifera* (MDR)	7.00 ± 0.00	0.00 ± 0.00	0.00 ± 0.00	7	0	9
43	*Proteus vulgaris* (MDR)	0.00 ± 0.00	0.00 ± 0.00	0.00 ± 0.00	11	9	11
44	*Streptococcus pneumoniae* (MDR)	0.00 ± 0.00	0.00 ± 0.00	0.00 ± 0.00	10	9	8
45	*Staphylococcus aureus* (MRSA)	12.00 ± 0.00	12.00 ± 0.00	13.00 ± 0.00	0	12	7
46	*Staphylococcus aureus* (MRSA + MDR)	10.00 ± 0.00	11.00 ± 0.00	13.00 ± 0.00	22	22	21
47	*Providencia rustigianii* (MDR)	7.00 ± 0.00	0.00 ± 0.00	0.00 ± 0.00	16	0	19
48	*Achromobacter* sp. (MDR)	9.00 ± 0.00	11.00 ± 0.00	13.00 ± 0.00	9	0	0

^1^ The data are given as the mean values of three replicates with standard errors.

**Table 2 microorganisms-11-00978-t002:** DPPH radical scavenging activity results for AFEt and ascorbic acid (%).

Concentrations (µg/mL)	AFEt (%)	Ascorbic Acid (%)
200.000	85.37	94.67
100.000	73.32	93.39
50.000	45.54	92.08
25.000	27.28	90.09
12.500	14.59	69.94
6.250	7.99	35.79
3.125	4.96	17.70
1.075	2.82	8.74

**Table 3 microorganisms-11-00978-t003:** The fold increases in biofilm formation in response to AFAq.

Microorganisms	Fold Increase
*E. coli* (CI)	1.58
*E. coli* ATCC 25922	2.63
*L. innocua* (FI)	1.24
*L. monocytogenes* ATCC 7644	1.76
*B. subtilis* DSMZ 1971	1.67

**Table 4 microorganisms-11-00978-t004:** GC-MS analysis of AFEt.

No	Retention Time	Chemical Structure ^1^	Compound Name ^1,2^	Formula ^1,2^	Molecular Weight (g/mol) ^1,2^	Area (%)	Known Activity
1	11.849	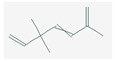	1,3,6-Heptatriene, 2,5,5-trimethyl	C_10_H_16_	136.23	2.96	-
2	12.303	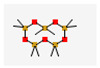	Decamethylcyclopentasiloxane	C_10_H_30_O_5_Si_5_	370.77	4.54	-
3	12.933	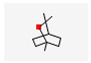	Eucalyptol	C_10_H_18_O	154.25	0.78	-
4	14.208	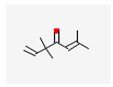	Artemisia ketone	C_10_H_16_O	152.23	19.41	Antimicrobial activity [22], antifungal activity [23]
5	19.301	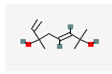	3,7-Octadiene-2,6-diol, 2,6-dimethyl	C_10_H_18_O_2_	170.25	1.94	-
6	22.060	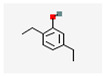	2,5-Diethylphenol	C_10_H_14_O	150.22	1.59	-
7	23.676	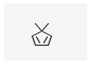	1,3-Cyclopentadiene, 5,5-dimethyl	C_7_H_10_	94.15	4.48	-
8	25.635	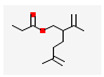	Lavandulyl propionate	C_13_H_22_O_2_	210.31	1.82	-
9	29.715	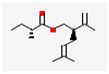	(R)-lavandulyl (R)-2-methylbutanoate	C_15_H_26_O_2_	238.37	4.86	-
10	31.789	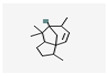	Di-epi-.alpha.-cedrene-(I)	C_15_H_24_	204.35	2.50	-
11	32.161	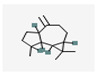	Aromadendrene	C_15_H_24_	204.35	5.11	Antimicrobial activity [24]
12	33.210	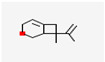	Exo-8-(2-Propenyl)-endo-8-methyl-3-oxabicyclo [4.2.0]oct-5-ene	C_11_H_16_O	164.24	2.98	-
13	33.330	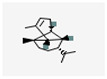	Copaene	C_15_H_24_	204.35	2.98	Antioxidant, antigenotoxic, and antiproliferative activities [25]
14	33.816	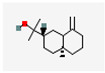	2-Naphthalenemethanol, decahydro-alpha,alpha,4a-trimethyl-8-methylene-, [2R-(2alpha,4aalpha,8abeta)]-	C_15_H_26_O	222.37	2.84	-
15	34.049	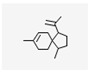	(1R,4S,5S)-1,8-Dimethyl-4-(prop-1-en-2-yl)spiro [4.5]dec-7-ene	C_15_H_24_	204.35	2.23	-
16	37.153	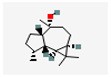	Ledol	C_15_H_26_O	222.37	1.31	Antimicrobial activity [26]
17	38.678	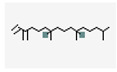	Neophytadiene	C_20_H_38_	278.50	0.65	Anti-inflammatory, analgesic, antipyretic, antioxidant, and antimicrobial activities [27]
18	41.944	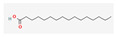	n-Hexadecanoic acid	C_16_H_32_O_2_	256.42	2.00	Anti-inflammatory [28], antioxidant, hypocholesterolemic, and antibacterial activities [29]
19	42.198	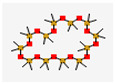	Cyclododecasiloxane, tetracosamethyl	C_24_H_72_O_12_Si_12_	889.84	1.08	-
20	54.739	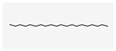	Eicosane	C_20_H_42_	282.54	1.98	-
21	59.453	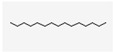	Pentadecane	C_15_H_32_	212.41	1.96	Antimicrobial activity [30]
22	64.305	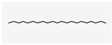	Heneicosane	C_21_H_44_	296.60	1.67	Antimicrobial activity [31]

^1^ https://pubchem.ncbi.nlm.nih.gov/; ^2^ https://webbook.nist.gov/chemistry/ accessed on 5 August 2022. “-”: no information.

## Data Availability

All the data generated or analyzed during this study are included in this published article.

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
