# Peer review of "Biological Activities and Biochemical Composition of Endemic Achillea fraasii"

_microorganisms, 2023, doi:10.3390/microorganisms11040978_

Round 1

Reviewer 1 Report (Previous Reviewer 1)

Thanks for attending all the suggestions. The manuscript has been significantly improved. I consider that the work has enough quality to be considered for publication in Microorganisms (MDPI).

Author Response

We want to express our gratitude for your suggestions and collaboration that helped in the advancement of the article during the review process.

Reviewer 2 Report (New Reviewer)

The paper describes the antimicrobial, antibiofilm and antioxidant activities of extracts of the plant, Achillea fraasii. The results are presented clearly and the paper is fairly straightforward. GC-Mass Spec analyses identify some compounds in the extracts and these are likely candidates for follow-on experiments.

I have a few short comments:

1: In Table 1, there is a general trend towards greater inhibition as the amount of extract added to the disc increases (form 50, to 100, to 200 ul). There are some exceptions though: For Escherichia coli ATCC 25922 , Providencia rustigianii Candida glabrata, and Escherichia coli at least, smaller, (or no) zones of inhibition are seen at higher levels of added extract. This needs some comment or explanation.

2. Lines 278-9; Please do not use your lab designations for bacteria/fungi. This is meaningless to the reader - identify the organism you're referring to.

3. Line 287. Provide some comment as to the meaning or significance of an 'EC50 value of AFEt was found to be 55.52'.

4. Line 293: Similarly, provide some comment as to the meaning or significance of a ' Pearson correlation coefficient = -0.9319768'. What are you concluding here ?

Author Response

1: In Table 1, there is a general trend toward greater inhibition as the amount of extract added to the disc increases (from 50 to 100, to 200 µL). There are some exceptions though: For Escherichia coli ATCC 25922, Providencia rustigianii, Candida glabrata, and Escherichia coli at least, smaller (or no) zones of inhibition are seen at higher levels of added extract. This needs some comment or explanation.
DONE
We have added a paragraph in the Discussion section addressing this issue and outlining the possible explanations.

2: Lines 278-9; Please do not use your lab designations for bacteria/fungi. This is meaningless to the reader - identify the organism you're referring to.
DONE
We have replaced the lab designations with numbers and added abbreviations in parentheses to avoid confusion regarding the strains.

3: Line 287. Provide some comment as to the meaning or significance of an 'EC50 value of AFEt was found to be 55.52'.
DONE
We have clarified the meaning of the EC50 value.

4: Line 293: Similarly, provide some comment as to the meaning or significance of a ' Pearson correlation coefficient = -0.9319768'. What are you concluding here?
DONE
We have provided a detailed explanation of the meaning and significance of the Pearson correlation coefficient.

Reviewer 3 Report (Previous Reviewer 3)

Although the recommendations were taken into account, the analysis of the results is superficial; I recommend going deeper into the findings, highlighting the results and making comparisons with the literature, even if they are of other species. 

Author Response

Although the recommendations were taken into account, the analysis of the results is superficial; I recommend going deeper into the findings, highlighting the results and making comparisons with the literature, even if they are of other species.
DONE
We have expanded the Discussion section by adding three paragraphs addressing the comparison of the effects of a closely related species, the importance of the strong effect observed for a specific strain, and the reduced effect observed for some strains despite increasing extract amounts. 

This manuscript is a resubmission of an earlier submission. The following is a list of the peer review reports and author responses from that submission.

Round 1

Reviewer 1 Report

The manuscript entitled “Biological Activities and Biochemical Composition of Endemic Achillea fraasii'', studies the A. fraasii ethanol and aqueous extracts were prepared, and antibacterial, antifungal, antioxidant, and antibiofilm activities and biochemical composition of the plant were determined. The authors founded that A. fraasii has antibacterial, antifungal, and antioxidant activities, but it does not have an antibiofilm activity; on the contrary, it has the effect of increasing  the biofilm structure. They should address the subject and critically review the information from the literature.

Minor suggestions:

The introduction does say little about the selected Achillea fraasii, why these were selected?

Please improve the abstract to cover the important topics reviewed and discussed in this article. The abstract is written in a way lacks logic. It should highlight the salient findings more critically;

The results have long paragraphs. I suggest reducing the size of the paragraphs. The results of this study are not fully explained therefore the interpretation of the results is very difficult. The author needs to provide the % increase or decrease rather than just writing ''significantly increased….'';

Authors should discuss the results integrally. The discussion is based on individual results. I suggest that integrating the results will give more value to the work. I suggest that you discuss by integrating all your results.

The conclusion is totally confusing. Re-write the conclusion! It needs to be much improved.

Author Response

The manuscript entitled “Biological Activities and Biochemical Composition of Endemic Achillea fraasii'', studies the A. fraasii ethanol and aqueous extracts were prepared, and antibacterial, antifungal, antioxidant, and antibiofilm activities and biochemical composition of the plant were determined. The authors founded that A. fraasii has antibacterial, antifungal, and antioxidant activities, but it does not have an antibiofilm activity; on the contrary, it has the effect of increasing the biofilm structure. They should address the subject and critically review the information from the literature.

DONE

Minor suggestions:

The introduction does say little about the selected Achillea fraasii, why these were selected?

DONE

Please improve the abstract to cover the important topics reviewed and discussed in this article. The abstract is written in a way lacks logic. It should highlight the salient findings more critically;

DONE

The results have long paragraphs. I suggest reducing the size of the paragraphs. The results of this study are not fully explained therefore the interpretation of the results is very difficult. The author needs to provide the % increase or decrease rather than just writing ''significantly increased….'';

DONE. THE CHANGES ARE GIVEN AS FOLD INCREASES.

Authors should discuss the results integrally. The discussion is based on individual results. I suggest that integrating the results will give more value to the work. I suggest that you discuss by integrating all your results.

DONE

The conclusion is totally confusing. Re-write the conclusion! It needs to be much improved.

DONE

Reviewer 2 Report

Major issues 

1) Introduction section is superficial, mostly presenting general knowledge in basic terms. The first few paragraphs should be rewritten, focusing more on information relevant to the current manuscript. Furthermore, English language should be corrected by a native speaker, as several phrases are hard to understand.

2) There is an unusual number of references provided for the Materials and Methods section. Considering that most methods are related to antibacterial activity determination, these citations do not have their place. Furthermore, a quarter of the 40 references are self-citations, which raises serious concerns regarding publication ethics.

3) References must be revised. Self-citations limited to articles relevant to the current research.

Minor issues

4) Line 19 - It is my understanding that the accepted usage is Türkiye, instead of Turkey. Why not use the appropriate diacritics ü? (both here and the affiliation page).

5) Line 38 - "synthetic products" would be a more appropriate expression.

6) Line 40-41 - Should be rephrased.

7) Line 93-94 - Please format the unit of measurement correctly.

8) Line 271-273: Considering that they are the same chemical compounds as previously tested (stated above) it doesn't seem relevant to test them again from AFEt isolates. An analysis on the synergistic effects of these compounds would be a more relevant proposal, to determine if AFEt has a more potent antibacterial effect than the sum of its components. If that would be the case, then the synergy between different compounds could be elucidated. The proposed studies from the conclusions are the ones I was referring to. These should be mentioned more clearly at the end of the discussions, and not repeated in the conclusions as well.

Author Response

Major issues 

1) Introduction section is superficial, mostly presenting general knowledge in basic terms. The first few paragraphs should be rewritten, focusing more on information relevant to the current manuscript.

DONE

Furthermore, English language should be corrected by a native speaker, as several phrases are hard to understand.

THE MANUSCRIPT WILL BE SENT TO A LANGUAGE EDITING.

2) There is an unusual number of references provided for the Materials and Methods section. Considering that most methods are related to antibacterial activity determination, these citations do not have their place. Furthermore, a quarter of the 40 references are self-citations, which raises serious concerns regarding publication ethics.

DONE

3) References must be revised. Self-citations limited to articles relevant to the current research.

DONE

Minor issues

4) Line 19 - It is my understanding that the accepted usage is Türkiye, instead of Turkey. Why not use the appropriate diacritics ü? (both here and the affiliation page).

DONE

5) Line 38 - "synthetic products" would be a more appropriate expression.

DONE

6) Line 40-41 - Should be rephrased.

DONE

7) Line 93-94 - Please format the unit of measurement correctly.

DONE

8) Line 271-273: Considering that they are the same chemical compounds as previously tested (stated above) it doesn't seem relevant to test them again from AFEt isolates. An analysis on the synergistic effects of these compounds would be a more relevant proposal, to determine if AFEt has a more potent antibacterial effect than the sum of its components. If that would be the case, then the synergy between different compounds could be elucidated. The proposed studies from the conclusions are the ones I was referring to. These should be mentioned more clearly at the end of the discussions, and not repeated in the conclusions as well.

DONE

Reviewer 3 Report

The idea of the article is good, however, in some points, it could be improved. Regarding the methodological proposal and the analysis of the results, it is mentioned that the antibacterial and antifungal activity and antioxidant activity tests were performed with the ethanolic extract, but for the antibiofilm test, they use the aqueous extract. Again, the characterization of the extract is performed for the ethanol extract and in the conclusion, they say: "It was found that A. fraasii has antibacterial, antifungal, and antioxidant activities, but it does not have an antibiofilm activity; on the contrary, it has the effect of increasing the biofilm structure". The two extracts are different, so different effects would be expected, the conclusion should clarify which extract is responsible for which activity.

And based on that: Why is the antibiofilm test done only for the water extract and not done for the ethanol extract? Moreover, the biochemical characterization is done only for the ethanol extract, all the tests are done with the ethanol extract, if biofilm formation is important to avoid infections, why do they change the extract?

·         The discussion is very superficial, only the important results are named, and the only explanation given is for the presence of compounds that have already been reported with these activities. What else can be said about the results found? About the differences between bacteria that are susceptible and those that are not. Between gram-positive and gram-negative bacteria? Could it be related to a possible mechanism by which the extract does these activities?

·         There is no comparison between the results and the literature and other ethanol extracts of other plants. Are the results comparable with what has been obtained? Are they better than what is being obtained to be able to think of a source of antibiotics?

·         In the extraction methodology, is the whole A. fraasii used? Or the 100 grams are from a specific part, flower, stem, or leaves? How long were the samples incubated in antimicrobial and antifungal? What concentration of the extract was used in the antioxidant assay? In the DPPH test, how is calculated the %? The equation is not shown. 

Author Response

The idea of the article is good, however, in some points, it could be improved. Regarding the methodological proposal and the analysis of the results, it is mentioned that the antibacterial and antifungal activity and antioxidant activity tests were performed with the ethanolic extract, but for the antibiofilm test, they use the aqueous extract. Again, the characterization of the extract is performed for the ethanol extract and in the conclusion, they say: "It was found that A. fraasii has antibacterial, antifungal, and antioxidant activities, but it does not have an antibiofilm activity; on the contrary, it has the effect of increasing the biofilm structure". The two extracts are different, so different effects would be expected, the conclusion should clarify which extract is responsible for which activity.

DONE

And based on that: Why is the antibiofilm test done only for the water extract and not done for the ethanol extract? Moreover, the biochemical characterization is done only for the ethanol extract, all the tests are done with the ethanol extract, if biofilm formation is important to avoid infections, why do they change the extract?

DUE TO THE LIMITATIONS OF METHODS USED IN ANTIMICROBIAL ACTIVITY AND ANTIBIOFILM ACTIVITY DIFFERENT EXTRACTS WERE USED. IF WATER EXTRACT WAS USED IN THE DISK DIFFUSION TEST, EVAPORATING WATER FROM DISKS WON’T BE EASY. AND IF ETHANOL EXTRACT WAS USED IN THE ANTIBIOFILM TEST, THE EXTRACT WILL CONTAIN ETHANOL WHICH COULD AFFECT THE GROWTH OF MICROORGANISMS.

  • The discussion is very superficial, only the important results are named, and the only explanation given is for the presence of compounds that have already been reported with these activities. What else can be said about the results found? About the differences between bacteria that are susceptible and those that are not. Between gram-positive and gram-negative bacteria? Could it be related to a possible mechanism by which the extract does these activities?

DONE

  • There is no comparison between the results and the literature and other ethanol extracts of other plants. Are the results comparable with what has been obtained? Are they better than what is being obtained to be able to think of a source of antibiotics?

 DONE

  • In the extraction methodology, is the whole A. fraasiiused? Or the 100 grams are from a specific part, flower, stem, or leaves?

DONE

How long were the samples incubated in antimicrobial and antifungal?

DONE

What concentration of the extract was used in the antioxidant assay?

DONE

In the DPPH test, how is calculated the %? The equation is not shown. 

DONE

Round 2

Reviewer 2 Report

Major issues 

1) Introduction section is superficial, mostly presenting general knowledge in basic terms. The first few paragraphs should be rewritten, focusing more on information relevant to the current manuscript.

DONE

 Reviewer's comment: The changes are adequate and acceptable.

Furthermore, English language should be corrected by a native speaker, as several phrases are hard to understand.

THE MANUSCRIPT WILL BE SENT TO A LANGUAGE EDITING.

  Reviewer's comment: I would suggest that in the future, language editing be performed prior to submitting the manuscript.

2) There is an unusual number of references provided for the Materials and Methods section. Considering that most methods are related to antibacterial activity determination, these citations do not have their place. Furthermore, a quarter of the 40 references are self-citations, which raises serious concerns regarding publication ethics.

DONE

 Reviewer's comment: As the current manuscript was marked as showing concerns regarding publication ethics, I evaluated carefully the references section. While the number of self-citation was indeed reduced (but there are still at least 6 or 7), there are still several issues regarding what and how the manuscript cites. Without detailing all discrepancies, here are some examples:

- Ref.2 used as source for WHO related information.

- Ref. 20 is used once, for a basic method description.

- Ref. 22 is used once, for a basic method description

- Ref. 25 used once, for a basic method description, for GC-MS conditions. The conditions are either standard (in which case no reference is needed) or they were developed for a previous study, in which case optimization / validation would have been required for the current samples.

- Ref. 26 is irrelevant.

3) References must be revised. Self-citations limited to articles relevant to the current research.

DONE

Reviewer's comment:  See above the comment regarding use of citations and references. While some measures have been taken to reduce self-citation, there are still some major concerns regarding this issue, which limits the merit of the manuscript.